# OrganoID: A versatile deep learning platform for tracking and analysis of single-organoid dynamics

Jonathan M. Matthews[1,2,3☯], Brooke Schuster[1,2,4☯], Sara Saheb Kashaf[1,2,3], Ping Liu[5], Rakefet Ben-Yishay[6,7], Dana Ishay-Ronen[6,7], Evgeny Izumchenko[8], Le Shen[9,10], Christopher R. Weber[10,11], Margaret Bielski[8], Sonia S. Kupfer[8], Mustafa Bilgic[5], Andrey Rzhetsky[2,8,12], Savaş Tay[1,2]*

**1** Pritzker School of Molecular Engineering, The University of Chicago, Chicago, Illinois, United States of America, **2** Institute for Genomics and Systems Biology, The University of Chicago, Chicago, Illinois, United States of America, **3** Pritzker School of Medicine, The University of Chicago, Chicago, Illinois, United States of America, **4** Department of Chemistry, The University of Chicago, Chicago, Illinois, United States of America, **5** Department of Computer Science, Illinois Institute of Technology, Chicago, Illinois, United States of America, **6** Institute of Oncology, Sheba Medical Center, Ramat-Gan, Israel, **7** Sackler Faculty of Medicine, Tel Aviv University, Tel Aviv, Israel, **8** Department of Medicine, The University of Chicago, Chicago, Illinois, United States of America, **9** Department of Pathology, The University of Chicago, Chicago, Illinois, United States of America, **10** Organoid and Primary Culture Research Core, The University of Chicago, Chicago, Illinois, United States of America, **11** Department of Surgery, The University of Chicago, Chicago, Illinois, United States of America, **12** Department of Human Genetics, The University of Chicago, Chicago, Illinois, United States of America

☯ These authors contributed equally to this work.
* tays@uchicago.edu

**Data Availability Statement:** We have released the OrganoID platform open-source and freely licensed on GitHub (https://github.com/jono-m/OrganoID). The repository includes all source code as well as

## Abstract

Organoids have immense potential as *ex vivo* disease models for drug discovery and personalized drug screening. Dynamic changes in individual organoid morphology, number, and size can indicate important drug responses. However, these metrics are difficult and labor-intensive to obtain for high-throughput image datasets. Here, we present OrganoID, a robust image analysis platform that automatically recognizes, labels, and tracks single organoids, pixel-by-pixel, in brightfield and phase-contrast microscopy experiments. The platform was trained on images of pancreatic cancer organoids and validated on separate images of pancreatic, lung, colon, and adenoid cystic carcinoma organoids, which showed excellent agreement with manual measurements of organoid count (95%) and size (97%) without any parameter adjustments. Single-organoid tracking accuracy remained above 89% over a four-day time-lapse microscopy study. Automated single-organoid morphology analysis of a chemotherapy dose-response experiment identified strong dose effect sizes on organoid circularity, solidity, and eccentricity. OrganoID enables straightforward, detailed, and accurate image analysis to accelerate the use of organoids in high-throughput, data-intensive biomedical applications.

the entire training and testing dataset, usage instructions, and scripts used for the examples presented in this paper. The network training module is also included on the repository to allow further model training to improve performance for any untested applications. The dataset is also available at https://osf.io/xmes4/.

**Funding:** This work is supported by NIH R01 GM127527 and P. G. Allen Distinguished Investigator Award (https://pgafamilyfoundation. org/) to S.T. The funders had no role in study design, data collection and analysis, decision to publish, or preparation of the manuscript.

**Competing interests:** The authors have declared that no competing interests exist.

## Author summary

A recent advance in biomedical research is the use of connective tissue gels to grow cells into microscopic structures, called *organoids*, that preserve and exhibit the physical and molecular traits of a particular organ. Organoids have enabled researchers to study complex phenomena, such as the beating of the heart or the folds of the intestines, and the effects of drugs on these properties. Changes in the size and shape of organoids are important indicators of drug response, but these are tedious to measure in large drug screening experiments, where thousands of microscopy images must be analyzed. We developed a software tool named *OrganoID* that automatically traces the exact shape of individual organoids in an image, even when multiple organoids are clumped together, and measures organoid changes over time. To show our tool in action, we used OrganoID to analyze pancreatic cancer organoids and their response to chemotherapy. We also showed that our tool can handle images of many different types of organoids, even those derived from mouse cells. With this software, researchers will be able to easily analyze immense quantities of organoid images in large-scale experiments to discover new drug treatments for a range of diseases.

This is a *PLOS Computational Biology* Software paper.

## Introduction

Organoids are multicellular three-dimensional (3D) structures derived from primary or stem cells that are embedded into a biological matrix to create an extracellular environment that provides structural support and key growth factors. Organoids closely recapitulate cellular heterogeneity, structural morphology, and organ-specific functions of a variety of tissues, which improves modeling of *in situ* biological phenomena compared to traditional monolayer cell cultures [1]. Live-cell imaging can reveal important features of organoid dynamics, such as growth, apoptosis/necrosis, movement, and drug responses, which can reflect physiological and pathological events such as organ development, function, infarction, cancer, and infection [2–7].

While organoids have been successfully used to investigate important phenomena that might be obscured in simpler models, their use in data-intensive applications, such as high-throughput screening, has been more difficult. A major challenge for organoid experiments is drug-response measurement and analysis, which must be performed for a large number of microscopy images. Image analysis is particularly difficult for organoid experiments due to the movement of organoids across focal planes and variability in organoid size and shape between different tissue types, within the same tissue type, and within the same single culture sample [8,9]. Several recently developed organoid platforms, while powerful, rely on per-experiment or per-image tuning of brightfield image analysis parameters [10–12], or require manual labeling of each image [7,13,14], which limits experiment reproducibility and scale. Organoids can be fluorescently labeled to aid in image segmentation and tracking, such as through genetic modification for fluorescent protein expression [15–17], cell fixation and staining [10,16], or the use of membrane-permeable dyes. However, these approaches may alter intrinsic cellular dynamics from original samples [18,19], limit measurements to endpoint assays, or cause cumulative toxicity through longer growth times and limited diffusion through the hydrogel

matrix [20]. There is a critical need for an automated image analysis tool that can robustly and reproducibly measure live-cell organoid responses in high-throughput experiments without the use of potentially toxic or confounding fluorescence techniques.

A number of software tools have been developed to automate the process of brightfield/ phase-contrast organoid image analysis. These platforms use conventional image processing methods, such as adaptive thresholding and mathematical morphology [21], or convolutional neural networks [22–24] to identify organoids in sequences of microscopy images. Despite their advantages, many existing platforms require cellular nuclei to be transgenically labeled [23], which increases experiment time and complexity and may modify cellular dynamics. Other existing platforms require manual tuning of parameters for each image [21], focus on single-timepoint analysis [24], only provide population-averaged (bulk) measurements without single-organoid resolution, or are limited to bounding-box detection [22], which fails to capture potentially useful morphological information at the individual organoid level. Changes in organoid shape, such as spiking or blebbing, can reveal important responses to external stimuli and might be missed with bounding-box measurements [25]. Many of these existing platforms were also developed for the analysis of organoids derived from a single type of tissue and for images obtained with one specific optical configuration, precluding their use across different experimental settings.

To address these challenges, we have developed a software platform, OrganoID, that can identify and track individual organoids in a population derived from a wide range of tissue types, pixel by pixel, in both brightfield and phase-contrast microscopy images and in time-lapse experiments. OrganoID consists of (i) a convolutional neural network, which detects organoids in microscopy images, (ii) an identification module, which resolves contours to label individual organoids, and (iii) a tracking module, which follows identified organoids in time-lapse imaging experiments. Most importantly, OrganoID was able to accurately segment and track a wide range of organoid types, including those derived from pancreatic ductal adenocarcinoma, adenoid cystic carcinoma, and lung and colon epithelia, without cell labeling or parameter tuning. The OrganoID software overcomes a major hurdle to organoid image analysis and supports wider integration of the organoid model into high-throughput applications.

## Design and implementation

### A convolutional neural network for pixel-by-pixel organoid detection

We developed a deep learning-based image analysis pipeline, OrganoID, that recognizes and tracks individual organoids, pixel by pixel, for bulk and single-organoid analysis in brightfield and phase-contrast microscopy images (**Fig 1A**). The platform employs a convolutional neural network to transform microscopy images into organoid probability images, where brightness values represent the network belief that an organoid is present at a given pixel. The network structure was derived from the widely successful *u-net* approach to image segmentation [26] (**Fig 1B**). The *u-net* approach first passes each image through a contracting series of multidimensional convolutions and maximum filters to extract a set of deep feature maps that describe the image at various levels of detail and contexts, such as edges, shapes, and textures. The feature maps are then passed through an expanding series of transposed convolutions, which learn to localize the features and assemble a final output.

The OrganoID neural network follows the *u-net* structure and was optimized to require far fewer feature channels than the original implementation, as a reduction in trainable model parameters limits overfitting and minimizes the amount of memory and computational power required to use the network in the final distribution [27]. The original *u-net* implementation uses 64 filters in the first layer, which results in quite a large number of trainable parameters

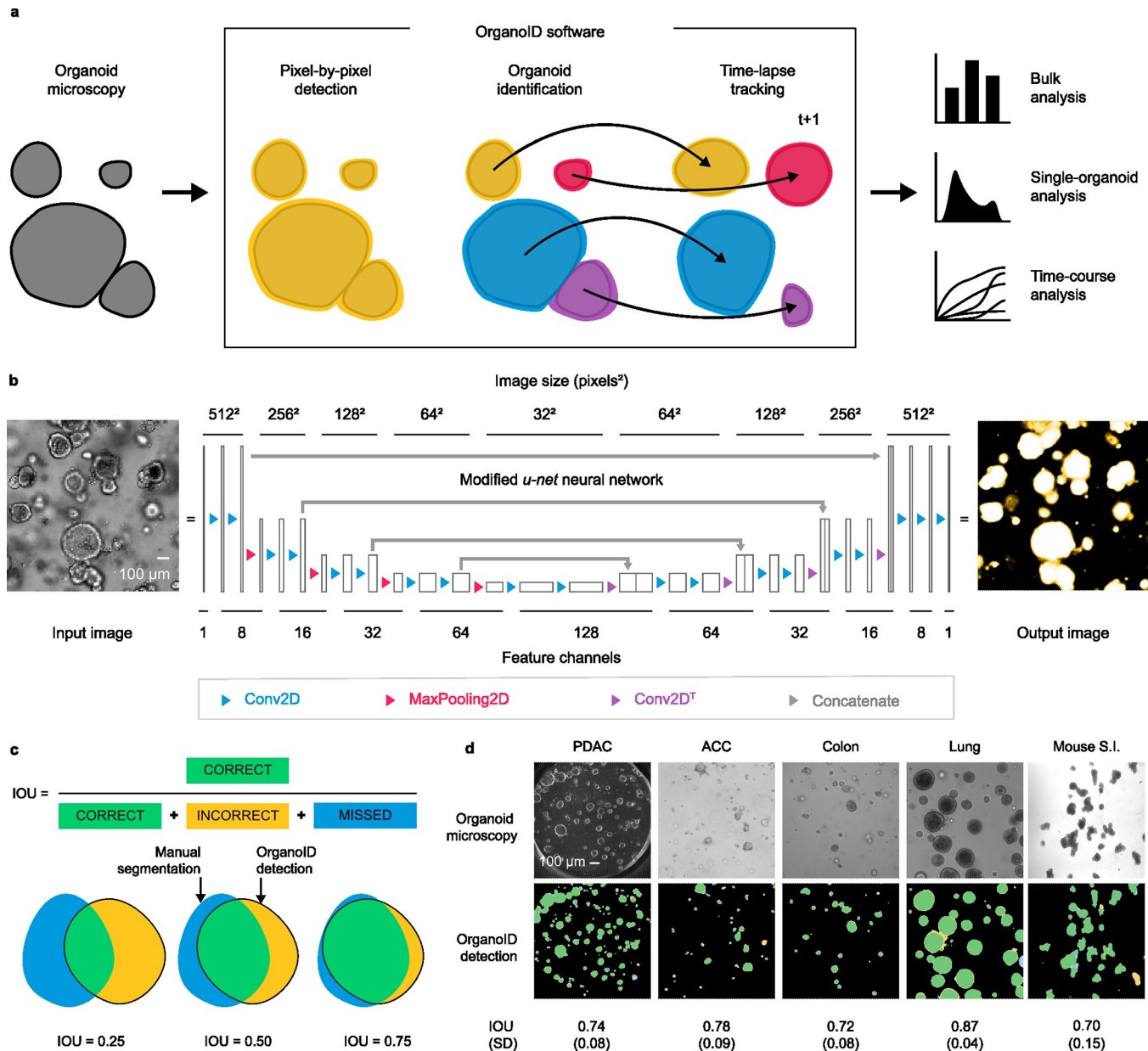

**Fig 1. Architecture and evaluation of the OrganoID platform and neural network. (a)** The OrganoID software automates robust analysis of organoid microscopy images. Contours are detected pixel by pixel and then separated into distinct organoids for bulk or single-organoid analysis. Identified organoids can also be tracked across time-lapse image sequences to follow responses over time. **(b)** Microscopy images are processed by a convolutional neural network to produce images that represent the probability that an organoid is present at each pixel. The network follows the *u-net* architecture, which applies a series of two-dimension convolutions, maximum filters, and image concatenations to extract and localize image features. Feature channel depths were minimized to limit overfitting and computational power required to use the tool. Scale bar 100 μm. **(c)** The intersection-over-union (IOU) metric, defined as the ratio of the number true positive pixels to the union of all positive pixels, was used to measure the quality of the neural network detections. To compute the IOU, pixels above 0.5 in the network prediction image were marked as positives. Examples of IOU values for several degrees of overlap are shown. **(d)** A representative set of images of different organoid types from the test dataset are shown with the corresponding OrganoID detections. Detections are overlaid on top of ground truth measurements and colored according to the schema in (c). The mean and standard deviation of IOU for images of each organoid type in the test dataset are also shown. Scale bar 100 μm.

across the full network (over 30 million) [26]. We sequentially reduced the number of filters in the first layer by powers of two to reach a minimal value that preserved performance on the validation dataset (validation cross-entropy loss was minimal at 0.088 for 8 filters in the first

layer, compared to 0.098 for 4 filters and 0.1 for 16 filters). The final OrganoID *u-net* structure uses only 8 filters in the first layer, which results in a network structure with less than 500,000 trainable parameters (a 98% reduction compared to the original implementation). All convolutional neurons were set to compute outputs with the exponential linear unit activation function, which has been previously shown to produce higher accuracy than the rectified linear unit (ReLU, used in the original *u-net* implementation) and avoid vanishing gradient problems during network training [28]. Neurons in the final convolutional layer were set with a sigmoid activation function to produce a normalized output that corresponds to the probability that an organoid is present at each pixel in the original image. All images are automatically contrasted and resized to 512x512 pixels before training and inference. Python was used for the entire OrganoID platform and Keras (a neural network interface for the TensorFlow software library) was used for network expression, training, and operation.

An original dataset of 66 brightfield and phase-contrast microscopy images of organoids were manually segmented to produce black-and-white ground truth images for network training (52 image pairs) and validation (14 image pairs) (**S1A Fig**). Each image featured 5 to 50 organoids derived from human pancreatic ductal adenocarcinoma (PDAC) samples from two different patients. Organoids were either grown on a standard tissue culture plate or on a microfluidic organoid platform [29]. To teach the network that segmentations should be independent of imaging orientation, field of view, lens distortion, and other potential sources of overfitting, the training dataset was processed with the Augmentor Python package [30] with random rotation, zoom, elastic distortion, and shear transformations to produce an augmented set of 2,000 images for training (**S1B Fig**). The neural network was trained on this dataset with the Adam stochastic optimization algorithm [31] at a learning rate of 0.001. Unweighted binary cross-entropy between predicted and ground-truth segmentations was used for the loss function. Layer weights were initialized with the He method [32]. An early stopping rule was used to halt training once performance on the validation dataset reached a minimum (i.e. once 10 epochs pass with no improvement in validation loss). The batch size was set to train on 8 augmented images for every round of backpropagation. Dropout regularization was introduced after all convolutions to randomly set 12.5% of neuron weights to zero after each batch. After each epoch, a copy of the model was saved for additional evaluation of the training process.

## Identification of individual organoids with diverse morphology and size

The convolutional neural network detects organoids in an image on a pixel-by-pixel basis, which can be used to measure bulk responses. For single-organoid analysis, pixels must be grouped together to identify individual organoids. This task is straightforward for isolated organoids, where all high-belief pixels in a cluster correspond to one organoid, but is more difficult for organoids that are in physical contact. To address this challenge, we developed an organoid separation pipeline (**Fig 2A**) that uses the raw network prediction image to group pixels into single-organoid clusters. Conventionally, neural network image segmentation methods set an absolute threshold on predicted pixels to produce a binary detection mask. This approach is effective but discards useful information about the strength of predictions. The segmentations that were used to train the neural network were produced with a 2-pixel boundary between organoids in contact that follows the contour of separation. As a result, the network predictions were marginally less confident about pixels near organoid boundaries (**Fig 2A**). We took advantage of this phenomenon to identify organoid contours with a Canny edge detector [33], an image transformation which (i) computes the partial derivative of pixel intensity to identify high-contrast areas, (ii) blurs the image to smooth noisy regions, and (iii)

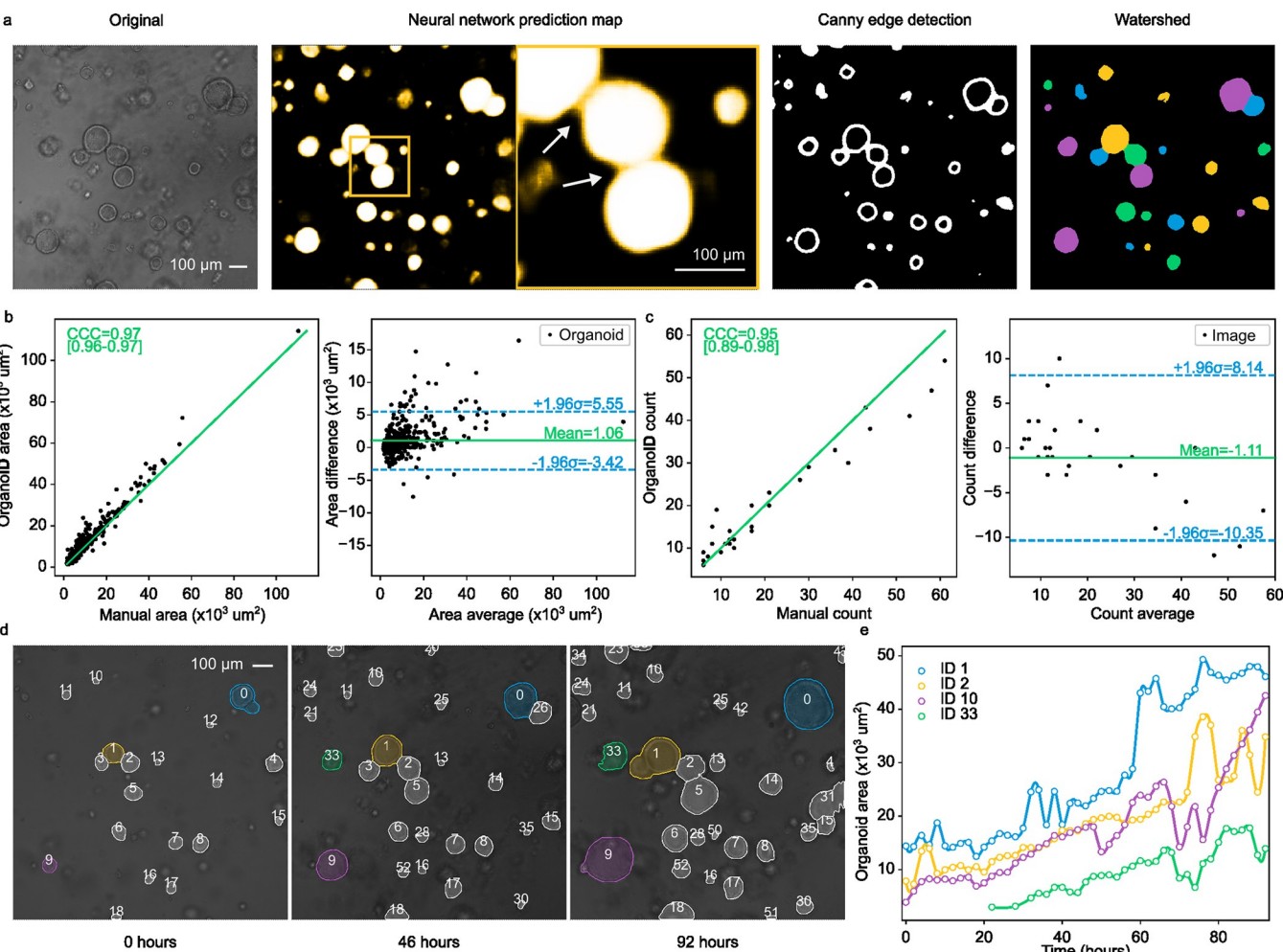

**Fig 2. (a)** OrganoID can identify individual organoid contours, including those in physical contact. An example image (left) is shown to demonstrate the steps of single-organoid identification. The neural network predictions (second-from-left) are observably less confident for pixels at organoid boundaries (enlarged view, indicated with white arrows), which enables edge detection with a Canny filter (second-from-right). Edges are used to identify organoid centers, which serve as basin initializers for a watershed transformation on the prediction image to produce a final single-organoid labeled image (right). **(b)** The identification pipeline was used to count the number of organoids in images from the test dataset. These counts were compared to the number of organoids in the corresponding manually segmented images. The concordance correlation coefficient (CCC) was computed to quantify measurement agreement (left). Bland-Altman analysis (right) demonstrates low measurement bias and limits of agreement. Black dots are test images. Green line in the left plot is $y = x$. **(c)** The area of each organoid in all test images was also measured manually and with OrganoID. Measurements were compared with CCC computation (left) and Bland-Altman analysis (right). Black dots are identified organoids. **(d)** Identified organoids in time-lapse microscopy images are matched across frames to generate single-organoid tracks and follow responses over time. Shown are images of three timepoints from an organoid culture experiment. **(e)** Automatically measured growth curves for a selected set of organoids from the experiment in (d).

applies a hysteresis-based threshold to identify locally strong edges. Edges are removed from the thresholded prediction image to mark the centers of each organoid. These centers are then used as initializer basins in a watershed transformation [34], an algorithm inspired by the filling of geological drainage basins that is used to segment contacting objects in an image. The image is further refined to remove organoids that may be partially out of the field-of-view or below a particular size threshold. The pipeline outputs a labeled image, where the pixels that represent an individual organoid are all set to a unique organoid ID number, which can be used for single-organoid analysis (**Fig 2A**).

The *scipy* [35] and *scikit-image* [36] packages were used to identify individual organoids from the network detection images. Detection images were thresholded at 0.5 and passed

through a morphological opening operation to remove weak and noisy predictions. Partial derivatives were computed with a Sobel filter, passed through a Gaussian filter ($\sigma = 2$), and converted to an edge mask with a hysteresis threshold filter ($T_{hi} = 0.05$, $T_{lo} = 0.005$), which was then was used to mark organoid centers. The watershed method was used to identify filled organoid contours, with the organoid centers as label initializers and the network prediction image as an inverted heightmap. Labeled organoids were morphologically filled and discarded if the total area was below 200 pixels. Images were morphologically processed with *scikit-image* to record properties of identified organoids.

### Automated organoid tracking in time-lapse microscopy experiments

OrganoID was also developed with a tracking module for longitudinal single-organoid analysis of time-course imaging experiments, where changes in various properties of individual organoids, such as size and shape, can be measured and followed over time. The central challenge for the tracking module is to match a detected organoid in a given image to the same organoid in a later image. The OrganoID tracking algorithm builds a cost-assignment matrix for each image in a time-lapse sequence, where each row corresponds to an organoid tracked from a previous image and each column corresponds to an organoid detected in the current image. Each matrix entry is the cost of assigning a given organoid detection in the current image to a detection in the previous image. The cost function was defined as the inverse of the number of shared pixels between the two organoids. As such, the assignment cost between two detections will be minimal for those that are of similar shape and in a similar location. The cost-assignment matrix is also padded with additional rows and columns to allow for "pseudo-assignments" that represent missing or newly detected organoids. Finally, the Munkres variant of the Hungarian algorithm [37] was used to minimize the cost-assignment matrix to find an optimal matching between organoid detections in the previous image and the detections in the current image. This approach produces organoid "tracks" for unique organoids in time-lapse images.

## Results

Network training halted after 37 epochs, when segmentation error (binary cross-entropy) on the validation dataset converged to a minimal value (**S1C Fig** and **S1 Video**). After hyperparameter tuning, the final model performance was assessed on a novel PDAC organoid testing dataset, previously unseen by the network. Performance was quantified with the intersection-over-union (IOU) metric, which is defined as the overlap between the predicted and actual organoid pixels divided by their union (**Fig 1C**). An IOU greater than 0.5 is generally considered to reflect a good prediction [38,39] and we chose this value to be the minimal benchmark for satisfactory model performance. All PDAC images passed the benchmark, with a mean IOU of 0.74 (SD = 0.081) (**S2 Fig**).

Because the network was trained and assessed solely with images of PDAC organoids, we were also curious to evaluate its capacity to generalize to organoids derived from non-PDAC tissues. Microscopy images of organoids derived from lung epithelia, colon epithelia, and salivary adenoid cystic carcinoma (ACC) were manually segmented (6 images for each tissue type). The OrganoID network passed the benchmark for all of these non-PDAC images, with a mean IOU of 0.79 (SD = 0.096) (**S2 Fig**). Additionally, 19 images of organoids derived from mouse small intestine were segmented and evaluated with the model and passed the benchmark with a mean IOU of 0.70 (SD = 0.15). These results support the potential of the OrganoID neural network to generalize to organoids from various tissues of origin without parameter tuning (**Fig 1D**). However, 3 images of mouse small intestinal organoids did not pass the benchmark, which suggests a limit to this generalizability. To demonstrate how

performance can be improved on novel datasets, a copy of the neural network was additionally trained on a subset of mouse small intestinal organoid images. This retrained network showed considerable improvement on a test dataset of mouse intestinal organoids with a mean IOU of 0.82 (SD = 0.09) while maintaining performance on the original test dataset with a mean IOU of 0.76 (SD = 0.10) (**S2 Fig**).

The network was also evaluated for appropriate exclusion of non-organoid technical artifacts. Air bubbles in culture media or gel matrix were rarely detected by OrganoID with a false positive rate of 4.2% (**S3A Fig**). We observed that OrganoID segmentations also avoided cellular debris or dust embedded into the gel, ignored chamber borders, and performed robustly across microscope resolutions, organoid concentrations, and organoid shapes (**S3B–S3G Fig**). OrganoID proved to be computationally efficient; each image was segmented in ~300 milliseconds on a laptop CPU (Intel i7-9750H, 2.6GHz) with less than 200 megabytes of RAM usage.

For quantitative validation of single-organoid analysis, OrganoID was used to count and measure the area of organoids in images from the PDAC and non-PDAC human testing datasets (a total of 28 images). These data were then compared to the number and area of organoids in the corresponding manually segmented images. Organoid counts agreed with a concordance correlation coefficient (CCC) of 0.95 [95% CI 0.89–0.98]. OrganoID, on average, detected 1.1 fewer organoids per image than manual segmentation. The limits of agreement between OrganoID and manual counts were between -10.35 and 8.14 organoids (**Fig 2B**). Organoid area comparison demonstrated a CCC of 0.97 [95% CI 0.96–0.97]. OrganoID area measurements were biased to be $1.06x10^3$ $\mu m^2$ larger per organoid with limits of agreement between $-3.42x10^3$ $\mu m^2$ and $5.55x10^3$ $\mu m^2$ (**Fig 2C**). Several additional morphological metrics were calculated, including circularity (the ratio of the organoid perimeter to the perimeter of a perfect circle), eccentricity (a measure of elliptical deviation from a circle), and solidity (the ratio of the organoid area to the area of its convex hull). These three metrics showed moderate concordance (0.52, 0.41, and 0.59 respectively) between automated and manual measurements (**S4 Fig**). Performance on brightfield microscopy images was higher than performance on phase-contrast microscopy images. Overall, the measurements produced by OrganoID were in considerable agreement with those obtained by hand in our test dataset, which supports the use of OrganoID for automated single-organoid analysis.

For validation of the organoid tracking module, microscopy images taken every 2 hours from a 92-hour organoid culture experiment were passed through the entire OrganoID pipeline to produce growth curves that followed single-organoid changes over time (**Fig 2D–2E**). The tracking step was also performed by hand to evaluate automated performance. OrganoID tracking maintained over 89% accuracy (defined as the fraction of identified organoids that were correctly matched at each time step) throughout the duration of the experiment (**S5 Fig and S2 Video**). Tracking accuracy was not affected when frames were discarded to simulate a 12-hour imaging interval, which demonstrates low susceptibility to imaging frequency.

## OrganoID normalizes single-organoid death responses over time

PDAC organoids were treated with serial dilutions of gemcitabine (3nM-1000nM), an FDA-approved chemotherapeutic agent commonly used to treat pancreatic cancer. Propidium iodide (PI), a fluorescent reporter of cellular necrosis, was also added to the culture media to monitor death responses. Cultures were then imaged every 4 hours over 72 hours and the produced brightfield images were processed with the OrganoID platform to identify organoids and analyze timelapse features (**Fig 3A**). At gemcitabine concentrations above 3 nM, the total organoid area increased for the first several hours, which reflected initial organoid growth, but then decreased to a value and at a rate inversely proportional to gemcitabine concentration

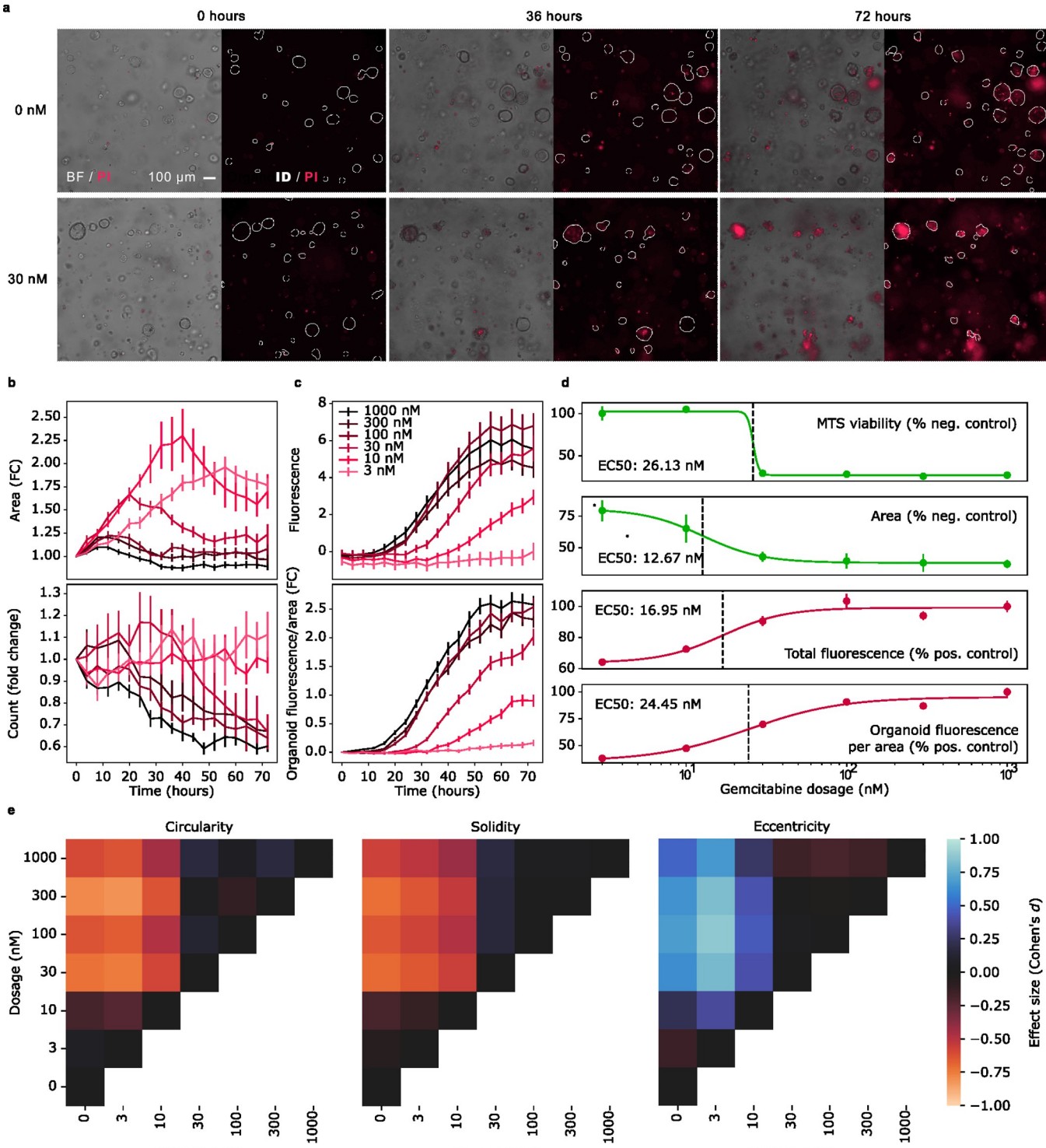

**Fig 3. OrganoID facilitates morphological analysis of a chemotherapeutic dose-response experiment. (a)** PDAC organoids were treated with a serial dilution of gemcitabine (3 nM to 1,000 nM) and imaged over 72 hours. Propidium iodide (PI) was used to fluorescently label dead organoids. Shown are representative brightfield images from three time points for control and 30 nM gemcitabine conditions. OrganoID was used to identify organoid contours, which are displayed on top of the PI fluorescence channel. **(b)** OrganoID measurements of fold change (FC) organoid area (top) and number of organoids (bottom) over time for each concentration of gemcitabine. Measurements were normalized to the initial timepoint. Error bars represent standard error of the mean (n = 6). **(c)** Total PI fluorescence intensity above control for each concentration of gemcitabine (top). The fluorescence intensity per area of each OrganoID-identified and tracked organoid were then normalized to the timepoint of first detection, which improved separation of responses to difference

concentrations of gemcitabine and reduced standard error across replicates (bottom). **(d)** An MTS assay was used as a gold standard measurement of viability to compute the half-maximal effective concentration ($EC_{50}$) at the end of the experiment. Organoid area, total fluorescence, and area-normalized fluorescence were also measured at endpoint to produce dose-response curves and estimate $EC_{50}$ without the need for live-cell staining. **(e)** Descriptive shape measurements, including circularity (the inverse ratio of organoid perimeter to the perimeter of a perfect circle with equivalent area), solidity (the ratio of organoid area to its convex hull), and eccentricity (elliptical deviation from a perfect circle), were computed for each organoid. The effect size (Cohen's *d* statistic) between each gemcitabine concentration is shown. Effect size is computed from dosages on the x-axis to dosages on the y-axis.

(**Fig 3B**). Identified organoid counts for gemcitabine concentrations above 10 nM also sharply decreased over time. The total fluorescence intensity of the PI signal increased over time to a value and at a rate proportional to gemcitabine concentration, however the response to 100 nM gemcitabine appeared to induce a stronger death signal than the response to 1000 nM gemcitabine (**Fig 3C**).

Measurements of death stains such as PI are typically normalized to a viability measurement that accounts for differences in the number and size of organoids between replicates that can compound over the duration of an experiment. We hypothesized that OrganoID could be used to measure morphological features indicative of organoid viability over time, which could then be used to normalize and decrease the variability of measurements across experimental replicates. Furthermore, organoid tracking with OrganoID would enable per-organoid measurement normalization, which we observed to significantly decrease the coefficient of variation of organoid area and fluorescence change across organoids imaged at the same time point and exposed to the same concentration of gemcitabine (**S6 Fig**). Normalization of PI intensity with tracked single-organoid area indeed increased the separation of responses between each treatment group over time, decreased standard error across replicates, and corrected the response discrepancy between the 100 nM and 1000 nM conditions (**Fig 3C**). We also compared dose-response measurements to an MTS proliferation assay ((3-(4,5-dimethylthiazol-2-yl)-5-(3-carboxymethoxyphenyl)-2-(4-sulfophenyl)-2H-tetrazolium), a gold-standard indicator of cell viability, at the experiment endpoint. The MTS assay determined the half-maximal effective concentration ($EC_{50}$) to be 26.13 nM of gemcitabine. However, total fluorescence and organoid area underestimated this value to be 16.95 nM and 12.67 nM, respectively. In contrast, the dose response of area-normalized fluorescence determined the $EC_{50}$ to be 24.45 nM, considerably closer to the MTS standard (**Fig 3D**).

## OrganoID uncovers morphology changes predictive of drug response

Changes in organoid morphology can indicate important phenotypic responses and state transitions. For example, some tumor organoid models grow into structures with invasive projections into the culture matrix, reflecting epithelial-mesenchymal transition; the addition of certain chemotherapeutic agents prevents development of these protrusions with minimal effects on overall organoid size [25]. These important responses can be investigated through single-organoid image analysis that captures the precise contour of each organoid. We used OrganoID to automatically profile the morphology of individual organoids across the gemcitabine dose-response experiment. Organoid circularity, solidity, and eccentricity showed sigmoidal dose responses and determined the endpoint $EC_{50}$ to be 28.47 nM, 27.05 nM, and 18.08 nM, respectively (**S7 Fig**). The effect size of gemcitabine concentration on these shape metrics was also computed with Cohen's *d* statistic, defined as the ratio of the difference in means to the pooled standard deviation. Gemcitabine dosage had a moderate to large endpoint effect on organoid circularity, solidity, and eccentricity across concentrations below and above the $EC_{50}$ (**Fig 3E**). Circularity and solidity decreased at higher concentrations, while eccentricity increased, reflecting disrupted organoid morphology. This worked example demonstrates

the advantages of OrganoID for automated bulk and single-organoid morphological analysis of time-course experiments without the need for live-cell fluorescence techniques.

## Availability and future directions

Organoids have revolutionized biomedical research through improved model representation of native tissues and organ systems. However, the field has yet to fully enter the high-throughput experimental space. A central bottleneck is the challenge of automated response measurement and analysis in large numbers of microscopy images. Organoids exhibit striking diversity in morphology and size and can move through their 3D environment into and out of the focal plane; current image processing tools have not quite been able to capture these aspects in a robust manner. We developed OrganoID to bridge this gap and automate the process of accurate pixel-by-pixel organoid identification and tracking over space and time.

Experimental replicates in organoid studies can differ in the number and distribution of sizes of organoids. This difference between identical conditions requires per-sample normalization of response measurements to baseline growth of organoid colonies. There are several commercially available live-cell assays that can facilitate normalization in 2D culture. However, these same assays have proven difficult for organoid use due to the production of toxic photobleaching byproducts, limited diffusion through the gel matrix, and nonspecific staining of the gel matrix that results in a considerable background signal. Another available option is to genetically modify each organoid sample to express fluorescent proteins; however, this method increases experimental time and complexity and may alter cellular dynamics from the original tissue sample. The OrganoID platform can be leveraged for accurate normalization of standard organoid assays without live-cell fluorescence methods. OrganoID is also uniquely useful for efficient quantification of single-organoid morphological features, such as circularity, solidity, and eccentricity, that can reflect important dynamic responses. In our gemcitabine dose-response experiment, we observed that PDAC organoid circularity and solidity decreased with an increase in gemcitabine dosage. This disturbance of organoid architecture is likely due to the interference of gemcitabine with RNA and DNA synthesis [40], which may in turn affect cell turnover and production of signaling and structural proteins.

In this work, we have also contributed a manually segmented organoid image dataset for use in other computational platforms. OrganoID has demonstrated compatibility with organoids of various sizes, shapes, and sample concentrations as well as various optical configurations. Most excitingly, the OrganoID model was trained and validated on images of PDAC organoids but still demonstrated excellent generalization to images of other types of organoids, including those derived from colon tissue, lung tissue, and adenoid cystic carcinoma.

OrganoID was trained with and tested on a diverse yet relatively small set of images. Despite the suggested generalizability of OrganoID to various samples and optical configurations, performance may still differ with other types of organoids with significantly different morphology. Performance on a dataset of mouse small intestinal organoids passed our benchmark on average, however there were several challenging images that were not processed adequately. We demonstrated further training on a handful of images of mouse small intestinal organoids to improve performance on such challenging datasets. As well, OrganoID can only detect and assign a single organoid to each pixel in an image. While the platform can appropriately identify contours of organoids in physical contact, it cannot distinguish organoids that overlap across the focal plane. These limitations could be overcome in future work with a network model that produces multiple outputs per pixel as well as additional validation, an expanded training dataset, and the use of multiple focal planes for image analysis.

We have released the OrganoID platform open-source and freely licensed on GitHub (https://github.com/jono-m/OrganoID). The repository includes all source code as well as usage instructions and scripts used for the examples presented in this paper. The network training module is also included on the repository to allow further model training to improve performance for any untested applications. The training and testing dataset is available at through the Open Science Framework (https://osf.io/xmes4/). Our image analysis platform serves as an important tool for the use of organoids as physiologically relevant models in high-throughput research. The platform can accurately capture detailed morphological measurements of individual organoids in live-cell microscopy experiments without the use of genetic modifications or potentially cytotoxic dyes. These metrics can reveal important organoid responses that might be otherwise obscured with the use of bounding-box tools. The ability of the platform to generalize to a range of organoid types without parameter tuning also reflects the potential for the platform to standardize morphological assay readouts and improve measurement reproducibility. OrganoID achieves comprehensive and expedient image analysis of organoid experiments to enable the broader use of organoids as tissue models for high-throughput investigations into biological systems.

## Supporting information

**S1 Fig. Network training performance. (a)** The network was trained on 66 manually labeled microscopy images of organoids derived from pancreatic ductal adenocarcinoma (PDAC) samples from two patients. Organoids were cultured in a well plate or microfluidic format and imaged through phase-contrast or brightfield microscopy. Images were then split into datasets for network training (80%) and validation (20%). **(b)** The 52 images in the training dataset were passed through a series of random transformations to produce an augmented dataset of 2,000 images. **(c)** Network training was stopped after 37 epochs, once a minimum binary cross-entropy loss on the validation dataset was reached. **(d)** The OrganoID neural network predicts the probability that an organoid is present at each pixel. Shown are network predictions produced by intermediate models at selected epochs through the training process.
(TIF)

**S2 Fig. Network test dataset performance.** A test set of images of organoids derived from human PDAC, salivary adenoid cystic carcinoma (ACC), colon epithelia, distal lung epithelia, and mouse small intestine were manually segmented to assess network performance (top). An IOU of 0.5 was set as a benchmark for a successful network prediction (dashed green line). All images of human organoids in the test set passed the benchmark, which demonstrates the capacity of the PDAC-trained network to generalize to other organoid types. Several of the mouse organoids did not pass the benchmark, and so the model was later retrained with part of this dataset included to demonstrate extensibility (bottom).
(TIF)

**S3 Fig. Exclusion of non-organoid artifacts.** OrganoID ignored bubbles **(a)**, debris **(c-d)**, and plate or microfluidic chamber borders **(f)** to accurately identify organoids that exhibit diverse morphology and sizes, even within a single sample **(b)**. OrganoID can also handle various optical configurations, including low-resolution or poorly-lit images **(e)**. Gel droplets can support densely-packed or isolated organoids, which can all be detected with OrganoID **(g)**.
(TIF)

**S4 Fig. Evaluation of organoid shape measurement.** OrganoID was used to measure organoid circularity (ratio of organoid area to the area of a perfect circle with equal perimeter), solidity (ratio of organoid area to area of the convex hull), and eccentricity (elliptical deviation

from a circle). These measurements were then compared to those from manual segmentation. The concordance correlation coefficient (CCC) was computed for all organoids, as well as for organoids imaged through phase contrast (PC, red) or brightfield (BF, blue) microscopy. For calculation of CCC for circularity and solidity (bounded to 0–1, with most values near 1), the data was first logit-transformed.
(TIF)

**S5 Fig. Tracking accuracy over time.** A time-lapse microscopy experiment was analyzed with OrganoID to identify organoids in each image. OrganoID was then used to match identified organoids across frames to build single-organoid tracks. The identified organoids were also matched by hand to assess tracking performance. Accuracy was defined the number of organoid track labels in agreement divided by the total number of organoids present at each frame.
(TIF)

**S6 Fig. Comparison of batch vs. tracked normalization.** In batch analysis, measurements of organoids exposed to the same gemcitabine dosage for the same duration were normalized to the average of organoids at t = 0. In tracked analysis, organoid measurements are instead normalized to each individual organoid measurement when initially detected by the tracking algorithm. The coefficient of variation (CV) was significantly lower with tracked analysis for change in fluorescence (Welch's t-test p = 2e-30), area (p = 2e-29) and perimeter (2e-63). CV was significantly higher with tracked analysis for eccentricity (p = 7e-25) and circularity (p = 2e-12).
(TIF)

**S7 Fig. Dose response of shape metrics.** Organoid circularity, solidity, and eccentricity were observed to follow sigmoidal dose responses to gemcitabine.
(TIF)

**S1 Video. Network training at the end of each epoch, representative image.**
(AVI)

**S2 Video. Organoid tracking video.**
(GIF)

**S1 Data. Numeric data used in all figures.**
(XLSX)

**S1 Methods. Further description of methods.**
(DOCX)

## Author Contributions

**Conceptualization:** Jonathan M. Matthews, Brooke Schuster, Sara Saheb Kashaf, Savaş Tay.

**Data curation:** Jonathan M. Matthews, Brooke Schuster, Rakefet Ben-Yishay, Dana Ishay-Ronen, Le Shen, Christopher R. Weber, Margaret Bielski, Sonia S. Kupfer.

**Formal analysis:** Jonathan M. Matthews, Brooke Schuster.

**Investigation:** Jonathan M. Matthews, Brooke Schuster, Sara Saheb Kashaf.

**Methodology:** Jonathan M. Matthews, Brooke Schuster, Sara Saheb Kashaf.

**Resources:** Rakefet Ben-Yishay, Dana Ishay-Ronen, Evgeny Izumchenko, Le Shen, Christopher R. Weber, Margaret Bielski, Sonia S. Kupfer, Savaş Tay.

**Software:** Jonathan M. Matthews, Sara Saheb Kashaf.

**Supervision:** Mustafa Bilgic, Andrey Rzhetsky, Savaş Tay.

**Validation:** Jonathan M. Matthews, Brooke Schuster, Ping Liu, Mustafa Bilgic, Andrey Rzhetsky.

**Visualization:** Jonathan M. Matthews.

**Writing – original draft:** Jonathan M. Matthews, Brooke Schuster, Mustafa Bilgic, Andrey Rzhetsky, Savaş Tay.

**Writing – review & editing:** Jonathan M. Matthews, Brooke Schuster, Sara Saheb Kashaf, Ping Liu, Rakefet Ben-Yishay, Dana Ishay-Ronen, Evgeny Izumchenko, Le Shen, Christopher R. Weber, Margaret Bielski, Sonia S. Kupfer, Mustafa Bilgic, Andrey Rzhetsky, Savaş Tay.

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
