## [Decision Letter · Decision Letter 0]

3 Jul 2022

Dear Professor Tay,

Thank you very much for submitting your manuscript "OrganoID: a versatile deep learning platform for tracking and analysis of single-organoid dynamics" for consideration at PLOS Computational Biology. As with all papers reviewed by the journal, your manuscript was reviewed by members of the editorial board and by several independent reviewers. The reviewers appreciated the attention to an important topic. Based on the reviews, we are likely to accept this manuscript for publication, providing that you modify the manuscript according to the review recommendations.

Sincerely,

Dina Schneidman

Software Editor

PLOS Computational Biology

[LINK]

Reviewer's Responses to Questions

**Comments to the Authors:**

Reviewer #1: High-confidence and robust methods for the analysis of easy-to-obtain brightfield images of organoid cultures are a central bottleneck for the high-throughput processing of clinically relevant in vitro drug-response screens and thus a timely problem.

Matthews and colleagues present a methodology to segment, track and analyze organoid shape from brightfield and phase-contrast microscopy images that can be potentially used for the high-throughput analysis of organoid screens. The heart of the analysis routine which they call OrganoID is a u-net-based neuronal network that was trained to segment organoids using a training data set of originally 66 images of pancreatic ductal adenocarcinoma (PDAC). The training data set was computationally increased by multiple rigid and elastic image transformations. The accuracy of the segmentation was compared to manually segmented ground-truth models and revealed a high IOU (0.74) for PDAC and of 0.79 for organoids from other tissues such as lung and colon as well as a low false positive rate (4.2%). To segment individual organoids the authors followed a classical Canny edge detection followed by watershed segmentation procedure. The number of automatically detected organoids was very close to manually identified organoids. Individual organoids were tracked over time with an estimated accuracy of 89%. Next the authors show a case study to quantify the response of PDAC organoids to the chemotherapeutic drug gemcitabine. Based on the segmented organoid masks they measure PI intensity as indicator for cell death and see a decrease in organoid area accompanied by an increase of PI with increasing drug concentration. Finally, the authors take circularity as one shape parameter and claim that it is predictive for a drug response.

Main points:

The presented analysis procedure works well on the test data set, and altogether appears to be promising. It is difficult, though, to really judge the robustness of the method due to the low number of test images. The authors themselves admit that the method was trained and tested on a small data set (66 PDAC and 18 other lines)– therefore it is difficult to argue that the method is ready to be used for big and diverse data sets. The authors should increase the test data set notably to deliver these arguments and find a way to quantify the robustness of the analysis pipeline for this bigger data set. Another point is that all organoid lines used in the manuscript are morphologically quite similar: round, cystic, rather homogenous. To argue for robustness in the recognition of different organoid shapes it would be more convincing to add organoid data e.g. of mouse intestinal organoids with recognizable differences in morphology characterized by protrusions, etc.

Additional points:

The authors segment, track and analyze individual organoids. This might be advantageous for many experimental designs and might moreover reduce the variance of the measured parameters. However, did the authors systematically compare batch analysis and individual (tracked) analysis? Is there a firm advantage of the individual tracking over a batch quantification for some features? Would for example geometrical features be less descriptive or informative at a batch quantification level of time snapshots as compared to individually tracked organoids? Does the variance change indeed?

How susceptible is the tracking algorithm for the time interval at which images are taken? For experimental reasons, the time interval might be as low a one or two images per day, would the tracking still work reasonably well and what would be the actual effect on tracking accuracy?

In Figure 3, the effect of gemcitabine should be shown as dose-response curves in addition to the temporal plots. Some of the dose-response curves in Supplementary Figure 5 should appear in the main figure. Does the cell area follow a dose-response curve as well? How does the EC50 measured based on PI or cell area measurements compare to the one determined by the MTS assay; do they match?

The analysis of cell shape parameters such as circularity, roundness, compactness, solidity, etc is more dependent on accurate pixel segmentation at the object’s circumference than of area. Filtering steps might smoothen the outlines of segmented organoids. How do shape parameters such as circularity compare between automatically and manually segmented organoids? For the analysis of circularity, the authors use phase-contrast microscopy images, whose analysis benefit from sharper contours. Does the shape quantification of brightfield image sources give similar results or are they more error-prone?

The authors do not explain well the selection of circularity as critical parameter for drug response. Did the authors also analyze other shape parameters in an unbiased way such as the ones mentioned above? I would suggest analyzing several parameters and find the most discriminative parameter. In general, the statement that circularity is a predictive parameter is not convincingly argued. The distribution of circularity and correlation with PI intensity seems minimal and the used statistical one-way ANOVA test to compare a few dozen cells should not be used. Instead, a test used for high effect size such as Cohen’s D should be used to test significance. Circularity should be shown in a dose-response curve.

Lastly, from the perspective of a potential user it would be interesting to know if the neuronal network can be readily extended by user-owned imaging data to improve the segmentation outcome and compensate for experimental variation due to differences in the specific equipment, imaging devices, etc. In case this can be done, it could be brought up in the discussion.

Reviewer #2: Heterogeneity in organoid morphology complicates high throughput, automated image analysis of multiple organoids growing within a substrate. Preparation of images for analysis by imaging software requires manual adjustment of focal plane and some “eyes-on” interpretation for each image is necessary. The authors have developed software that can monitor individual organoids representing any tissue type and provide morphometric data. The major accomplishment is the ability to distinguish multiple organoids within a cluster, and track and provide data on each. The strategy is label-free and non-destructive, enabling longitudinal studies. Overall, the manuscript is clear, concise, and well written. Impact is judged to be high, based on the tremendous need for automated tracking and characterization of individual human tissue derived organoids within a population of such organoids.

The technique described leverages artificial intelligence technology, of which, this reviewer has no specific expertise. However, the challenges of automated differentiation and tracking of overlapping, clonally expanding cell populations in microscopic images is well understood by many of us in the 3D in vitro modeling community.

Major strengths

Validation of an AI algorithm for discriminating, and tracking individual organoids within a population, over time, and in a label-free manner.

Ability to integrate morphometric analyses that provide some degree of phenotypic characterization. These capabilities were demonstrated through studies involving the determination of chemotherapeutic agent effectiveness on a pancreatic tumor organoid population. These studies did include propidium iodide, as training data. These experiments were then extended to label-free evaluation of organoid health in label-free studies. It is obvious that further algorithm training may expand significantly on these capabilities.

Overall, the manuscript is clear, concise, and well written. It was a pleasure to read and would be easily understood by someone with either a computer science or biomedical background.

Impact is judged to be high, based on the tremendous need for automated tracking and characterization of individual human tissue derived organoids within a population of such organoids.

Minor criticism

As stated above, the manuscript is easily followed by a reader with a life sciences background. That being said, this reviewer was forced to Google several terms to understand the significance. A few extra sentences would have made this task unnecessary for grasping the significance of sentences such as, “We took advantage of this phenomenon to identify and separate organoid contours with a modified Canny edge detector (to define organoid boarders) and a watershed transformation (to differentiate individual organoid within an overlapping cluster).

In summary, the manuscript describes an AI algorithm that can differentiate and track multiple individual organoids within a population that has a significant degree of overlap. This capability may be further enhanced through continued training that may provide phenotypic characterization based on morphometric parameters. The information is expertly communicated and the impact for this work is deemed high.

Reviewer #3: In this manuscript, the authors present their findings of a machine-learning-based image analysis model, OrganoID. The authors have developed a novel platform in OrganoID that can be used to monitor drug responses by tracking organoid dynamics via deep learning. The authors have emphasized the importance of studying organoids, highlighted the limitations of the existing approach, and clearly stated the motivation for an automated image analysis tool. Furthermore, the greatest strength of this model is that it can be applied to a variety of tissues. Overall the paper is very well written especially the results section which has been broken down under further subheadings to make the paper easily comprehensible. A few revisions and clarifications are recommended to make the paper more robust.

**Have the authors made all data and (if applicable) computational code underlying the findings in their manuscript fully available?**

Reviewer #1: Yes

Reviewer #2: Yes

Reviewer #3: Yes

PLOS authors have the option to publish the peer review history of their article (what does this mean?). If published, this will include your full peer review and any attached files.

Reviewer #1: No

Reviewer #2: No

Reviewer #3: No

Figure Files:

Data Requirements:

Reproducibility:

References:

---

## [Decision Letter · Decision Letter 1]

18 Sep 2022

Dear Professor Tay,

We are pleased to inform you that your manuscript 'OrganoID: a versatile deep learning platform for tracking and analysis of single-organoid dynamics' has been provisionally accepted for publication in PLOS Computational Biology.

Best regards,

Dina Schneidman

Software Editor

PLOS Computational Biology

Reviewer's Responses to Questions

**Comments to the Authors:**

Reviewer #1: The authors have adequately addressed my comments and major concerns. While the image set remains relatively small, I appreciate the addition of the mouse intestinal organoid images with more complex morphology to the test dataset. The additional quality assessments and controls will allow the readers to evaluate the strength and limitations of this new method.

The deposited instruction for model retraining to meet potential user-specific requirements is a valuable addition to the manuscript.

Reviewer #3: In this manuscript, the authors present their findings of a machine-learning-based image analysis

model, OrganoID. The authors have developed a novel platform in OrganoID that can be used to

monitor drug responses by tracking organoid dynamics via deep learning. The authors have

emphasized the importance of studying organoids, highlighted the limitations of the existing

approach, and clearly stated the motivation for an automated image analysis tool. Furthermore,

the greatest strength of this model is that it can be applied to a variety of tissues. The authors

have responded adequately to the reviewer's comments.

**Have the authors made all data and (if applicable) computational code underlying the findings in their manuscript fully available?**

Reviewer #1: Yes

Reviewer #3: Yes

PLOS authors have the option to publish the peer review history of their article (what does this mean?). If published, this will include your full peer review and any attached files.

Reviewer #1: No

Reviewer #3: No

---

## [Editor Report · Acceptance letter]

26 Oct 2022

PCOMPBIOL-D-22-00656R1 

OrganoID: a versatile deep learning platform for tracking and analysis of single-organoid dynamics

Dear Dr Tay,

I am pleased to inform you that your manuscript has been formally accepted for publication in PLOS Computational Biology. Your manuscript is now with our production department and you will be notified of the publication date in due course.

With kind regards,

Zsuzsanna Gémesi
